# Effects of C-Peptide on Dexamethasone-Induced In Vitro and In Vivo Models as a Potential Therapeutic Agent for Muscle Atrophy

**DOI:** 10.3390/ijms242015433

**Published:** 2023-10-21

**Authors:** Jinjoo Kim, Youngmo Yang, Eunwon Choi, Sumin Lee, Jiyoung Choi

**Affiliations:** 1Department of Food and Nutrition, College of Natural Science and Public Health and Safety, Chosun University, Gwangju 61452, Republic of Korea; jinjookim99@naver.com (J.K.); cyw0808@naver.com (E.C.); foodnutritionlee@naver.com (S.L.); 2Department of Pharmacy, College of Pharmacy, Chosun University, Gwangju 61452, Republic of Korea; yyang@chosun.ac.kr

**Keywords:** human C-peptide, dexamethasone, muscle atrophy, C2C12, skeletal muscle

## Abstract

This study aimed to investigate the effects of C-peptide on C2C12 myotubes and a mouse model. Both in vitro and in vivo experiments were conducted to elucidate the role of C-peptide in muscle atrophy. Various concentrations (0, 0.01, 0.1, 1, 10, and 100 nM) of C-peptide were used on the differentiated C2C12 myotubes with or without dexamethasone (DEX). C57BL/6J mice were administered with C-peptide and DEX for 8 days, followed by C-peptide treatment for 12 days. Compared to the DEX group, C-peptide increased the fusion and differentiation indices and suppressed atrophic factor expression in C2C12 myotubes. However, 100 nM C-peptide decreased the fusion and differentiation indices and increased atrophic factor expression regardless of DEX treatment. In C57BL/6J mice, DEX + C-peptide co-treatment significantly attenuated the body and muscle weight loss and improved the grip strength and cross-sectional area of the gastrocnemius (Gas) and quadriceps (Quad) muscles. C-peptide downregulated the mRNA and protein levels of muscle degradation-related markers, particularly Atrogin-1, in Gas and Quad muscles. This study underscores the potential of C-peptides in mitigating muscle weight reduction and preserving muscle function during muscle atrophy via molecular regulation. In addition, the work presents basic data for future studies on the effect of C-peptide on diabetic muscular dystrophy.

## 1. Introduction

Skeletal muscle is one of three types of muscles that make up the human body, together with the cardiac and visceral muscles. It constitutes approximately 40% of the entire body mass and encompasses 50–75% of body protein content [1]. Skeletal muscles perform multifaceted functions, including the maintenance of body posture, generation of movement, and regulation of body temperature, and act as pivotal metabolic hubs for glucose, lipids, and amino acids [2]. The balance between muscle protein synthesis and degradation, which is modulated by various signaling pathways, regulates muscle weight and function [3]. However, if by some mechanism the rate of muscle protein degradation exceeds that of protein synthesis, muscle weight, strength, and endurance decrease, leading to muscle atrophy [4]. Muscle atrophy is caused by various factors, including starvation, aging, immobility, cachexia, acute injuries, and chronic diseases, such as heart failure, chronic kidney disease, and obstructive pulmonary disease [5,6,7]. When muscle atrophy is induced, physical activity is restricted owing to muscle weakness, which subsequently leads to a vicious cycle of muscle loss. However, there is no cure for muscle atrophy, necessitating the development of novel preventive and therapeutic agents for this illness. The major muscle control mechanisms include the ubiquitin–proteasome system, insulin growth factor-1 (IGF-1)/phosphoinositide 3-kinase (PI3K)/protein kinase B (Akt), and autophagy pathways [8]. Within the ubiquitin–proteasome system, target proteins are marked by ubiquitin and subsequently degraded into amino acids by the proteasome [9]. During this process, the E3 ubiquitin ligase aids in the transfer of ubiquitin to the target protein. Consequently, muscle-specific E3 ubiquitin ligases, namely Muscle RING-finger protein-1 (MuRF1) and Atrogin-1, elicit augmented muscle protein degradation upon increased expression [10]. IGF-1 induces the activation of PI3K, amplifying the phosphorylation of Akt [11]. This leads to the activation of the mammalian target of rapamycin (mTOR), which ultimately upregulates the phosphorylation of the factors ribosomal protein S6 kinase B1 (P70 S6K1) and eukaryotic initiation factor 4E-binding protein 1 (4EBP1), which play pivotal roles in muscle protein synthesis [12]. Autophagy is an intricate process responsible for the degradation of organelles, macromolecules, and proteins, followed by the recycling of the resultant degradation byproducts, thereby constituting a crucial mechanism for cellular sustenance and upkeep [13]. However, deviations from optimal levels of autophagic activity within skeletal muscles, manifesting as either excessive or insufficient autophagy, can induce apoptosis and contribute to the development of muscle atrophy [14]. Dexamethasone (DEX), widely used to induce muscle atrophy, is a glucocorticoid-based compound that induces muscle atrophy by regulating the IGF-1/PI3K/Akt pathway [15]. Glucocorticoids, including DEX, exert an inhibitory influence on protein synthesis by attenuating mTOR activity, leading to the dephosphorylation of P70S6K1 and promotion of 4EBP1. In addition, glucocorticoids promote the dephosphorylation of forkhead box protein O (FoxO), resulting in the upregulation of MuRF1 and Atrogin-1, both pivotal factors contributing to skeletal muscle atrophy [16]. This shows similarities to the mechanism of muscular dystrophy caused by various factors mentioned earlier [17].

C-peptide is a connecting peptide that links the A and B chains of insulin, which consist of 21 and 30 amino acids, respectively. It is generated in equimolar amounts to insulin in pancreatic beta cells [18,19]. The half-life of C-peptide is 20–30 min, which is longer than that of insulin, whose half-life is only 3–5 min [20]. Thus, it has been used as a marker to evaluate pancreatic beta cell function [21]. Recent studies have revealed the various effects of C-peptide on diabetic complications [22]. The effects of C-peptide reported to date include increased intracellular calcium levels [23], activation of MAPK signaling [24], and stimulation of Na^+^/K^+^-ATPase activity [25,26]. In addition, the administration of C-peptide in hyperglycemic conditions due to pancreatic β-cell dysfunction has been reported to be effective in preventing or improving various diabetic complications, such as wound healing [27], muscle blood flow [28], neuropathy [29], kidney disease [30], and myocardial vasodilation [31]. The efficacy of C-peptide on muscle has been recently investigated [32]. Nevertheless, there is little investigation in the scientific literature on the specific role of C-peptide in muscle atrophy. This scarcity of research provides an impetus for the present investigation, which is grounded in earlier studies revealing the involvement of C-peptide in the activation of the PI3K pathway [33].

In this study, C2C12 mouse myoblast cell lines were used to evaluate the efficacy of various concentrations of C-peptide. In addition, animal experiments were performed to determine the molecular mechanism of C-peptide in the skeletal muscle after the co-administration of DEX and C-peptide.

## 2. Results

### 2.1. Effects of Various Concentrations of C-Peptide on C2C12 Myotubes in DEX-Induced Muscle Atrophy

To evaluate the effects of C-peptide on DEX-induced muscle atrophy, C2C12 myotubes were treated with 1 nM DEX for 5 h, followed by treatment with various concentrations of C-peptide (0.01, 0.1, 1, 10, and 100 nM) for an additional 5 h interval. Immunofluorescence staining showed that the myosin heavy chain (MHC)-positive area was noticeably decreased in the DEX group compared to the differentiation media (DM) group, resulting in a significant decrease in both the differentiation and fusion indices. The fusion index was calculated as the percentage of nuclei in MHC-positive myotubes (≥2 nuclei) in the total nuclei within the MHC-positive myotubes. In the DEX + C-peptide 1 nM (DEX + CP1) group, the decrease in DEX-induced MHC expression was hindered, leading to a significant increase in both the differentiation and fusion indices compared with the DEX group. However, in the group treated with 100 nM C-peptide this ameliorative effect was less pronounced, and the augmentation in the fusion index did not reach statistical significance compared to that in the DEX group (Figure 1A,B). Moreover, Western blotting analysis targeting MHC revealed that C-peptide within the range of 0.01–10 nM ameliorated the DEX-induced decline in MHC expression. Interestingly, Western blotting analysis revealed that 100 nM of C-peptide decreased the MHC expression, irrespective of DEX treatment (Figure 1C).

### 2.2. C-Peptide Co-Treatment Inhibits DEX-Induced Muscle Atrophy

To assess the onset of muscle atrophy in the C57BL/6J mice used in this study, we analyzed their body weight and grip strength. Body weight was significantly lower in the DEX group than in the control (CON) group from day 4 to the end of the experiment. However, co-treatment with C-peptide and DEX (DEX + CP group) significantly inhibited the weight loss induced by DEX from day 8 of the experiment, resulting in an improvement in final body weight and body weight gain (Figure 2A). In the DEX group, grip strength exhibited a marked reduction relative to that in the CON group. Intriguingly, DEX + CP group showed a tendency for increased grip strength compared to the DEX group (*p* = 0.0529; Figure 2B).

Major tissue weight measurements among each group revealed no significant differences in both absolute and relative weights of the gastrocnemius (Gas) muscle between the CON and DEX groups. In contrast, the DEX + CP group showed a significant increase in these parameters compared to the DEX group (Figure 2C). Furthermore, quadricep (Quad) muscle weight was significantly lower in the DEX group compared with the CON group. It was significantly increased in the DEX + CP group compared with the DEX group. When normalized by body weight, there was no significant difference in Quad muscle weight between the groups (Figure 2D). Similarly, no statistically significant differences were observed between the groups in the absolute and relative weights of the tibialis anterior (TA) muscle (Figure 2E). Both the weights of the epididymal adipose tissue and liver significantly decreased in the DEX group compared to those in the CON group, and significantly increased in the DEX + CP group compared to that in the DEX group (Figure 2F,G). Analysis of serum C-peptide concentration in mice revealed no difference in the serum C-peptide concentration between the DEX + CP and the two groups not administered with C-peptide, as well as between the CON and DEX groups (Figure 2H).

### 2.3. C-Peptide Attenuates the Reduction in Muscle Fiber Size in Mice with Muscle Atrophy

We evaluated the effect of C-peptide treatment on muscle fiber size by analyzing sections of the Gas and Quad muscles stained with hematoxylin-eosin (H&E). Regarding the cross-sectional area (CSA) measurements of the Gas muscle, the CON group showed a fiber size distribution at approximately 37.2% within the range of 1500–2000 µm^2^. Upon DEX treatment, there was a notable shift, with the range of 1000–1500 μm^2^ encompassing the majority at 30.1%. This change indicated an increase in the CSA in a smaller size range than that in the CON group. The DEX + CP group displayed a fiber size distribution of approximately 32.5% within the of 2000–2500 μm^2^, showing changes compared with the DEX group (Figure 3A,B). The average of CSA also significantly decreased in the DEX group compared to the CON group, and significantly increased in the DEX + CP group compared to the DEX group (Figure 3C).

In the Quad muscle, the CON group showed CSA of 18.8% in the range of 2500–3000 μm^2^, whereas the DEX group, showed a fiber size distribution of approximately 26.2% in the range of 500–1000 µm^2^. The DEX + CP group exhibited 18.0% of the 2500–3000 μm^2^ range in CSA, showing a similar distribution to the CON group. (Figure 3D,E). The mean of CSA also significantly decreased in the DEX group compared to the CON group and significantly increased in the DEX + CP group compared with the DEX group (Figure 3F).

### 2.4. C-Peptide Regulates the mRNA and Protein Levels in Gastrocnemius Muscles

In the Gas muscle, the expression levels of *Foxo1* and *Trim63* showed no notable differences among the experimental groups. In contrast, the expression levels of *Klf15* and *Fbxo32* were significantly lower in the DEX + CP group than in the CON and DEX groups. *Foxo3a* showed a tendency to decrease in the DEX group compared to the CON group and showed a significant decrease in the DEX + CP group compared to the CON and DEX groups. Additionally, *Trim63* expression was significantly decreased in the DEX + CP group in comparison to the CON group (Figure 4A). On the other hand, the expression levels of *Lc3b* and *Sqstm1*, which are autophagy-related genes, did not differ among groups (Figure 4B).

Western blotting analysis of the protein expression levels of p-P70 S6K1/P70 S6K1 and p-4EBP1/4EBP1 in Gas muscles showed no differences among the groups (Figure 4C). KLF15, p-Akt/Akt, and MuRF1 levels did not show differences among the groups; however, the expression of p-FoxO3a/FoxO3a increased more in the DEX + CP group than in the DEX group (*p* = 0.0583). Atrogin-1 levels were significantly lower in the DEX + CP group than in the CON and DEX groups (Figure 4D). Meanwhile, LC3B and p62 levels did not differ among the groups (Figure 4E).

### 2.5. C-Peptide Modulates the mRNA and Protein Levels in Quadriceps Muscles

mRNA expression analysis of Quad muscles revealed that the expression levels of *Klf15*, *Foxo1*, *Foxo3a*, and *Trim63* did not show any differences among the groups. *Fbxo32* expression was significantly decreased in the DEX + CP group compared to the CON group, and tended to decrease compared to the DEX group (*p* = 0.0829; Figure 5A). On the other hand, expression levels of autophagy-related genes, *Lc3b* and *Sqstm1*, showed no statistically significant differences among the groups (Figure 5B).

Protein expression analysis of the Quad muscle revealed that the expression of p-P70 S6K1/P70 S6K1 and p-4EBP1/4EBP1 did not differ between groups (Figure 5C). KLF15 and p-Akt/Akt levels did not differ between the groups. However, the phosphorylation of FoxO3a demonstrated a substantial reduction in the DEX group relative to the CON group, whereas there was a significant increase in the DEX + CP group in comparison to the DEX group. MuRF1 expression increased in the DEX group compared to the CON group and significantly decreased in the DEX + CP group compared to that in the DEX group. Furthermore, Atrogin-1 expression was significantly decreased in the DEX + CP group compared with that in the CON and DEX groups (Figure 5D). The autophagy-related markers LC3B and p62 showed no differences among groups (Figure 5E).

## 3. Discussion

Skeletal muscle is one of the tissues that make up our bodies and perform physical movements while maintaining body temperature [1,2,3]. As a major energy metabolism tissue in the human body, muscle loss and dysfunction significantly reduce quality of life. Muscle atrophy can occur not only due to starvation and aging but also as a side effect of sudden injury, medication, and various chronic illnesses resulting from an imbalance in muscle protein synthesis and degradation [5,6,7]. The mechanisms that regulate muscle synthesis and degradation include the ubiquitin–proteasome system, the IGF-1/PI3K/Akt pathway, and autophagy [8,9,10,11,12]. Muscle atrophy has not yet been treated, and the development of substances to delay or improve muscle atrophy is necessary.

C-peptide, the connecting peptide linking proinsulin, has been reported to have various physiological and biological activities in recent studies [18,19,20,21,22]. Particularly, based on previous studies regarding the PI3K activation effect of C-peptide [33], we hypothesized that C-peptide may have a beneficial effect on muscle atrophy by activating the IGF-1/PI3K/Akt pathway. Based on these hypotheses, this study was conducted to investigate the concentration-dependent physiological activity of C-peptide and to identify the muscle control mechanism in a DEX-induced muscle atrophy model.

In numerous cell experiments related to muscle atrophy, DEX-treated C2C12 myotubes demonstrated significant decreases in cell viability, myotube diameter, MHC staining area, and fusion index due to muscle atrophy [15,16,34]. In this study, when DEX-induced atrophic C2C12 myotubes were treated with 1 nM C-peptide, increases in the MHC staining area, differentiation index, and fusion index were observed. However, treatment with 100 nM C-peptide resulted in a reduction in the effect, showing no increase in the fusion index compared to the DEX group (Figure 1A,B). Western blotting analysis showed that MHC expression levels were decreased in cells treated with 100 nM C-peptide, irrespective of DEX treatment (Figure 1C). These findings suggest that at the cellular level, C-peptide exhibits anti-atrophic efficacy at concentrations below 10 nM, but unexpectedly induces muscle atrophy at higher concentrations, such as 100 nM. The reversal of muscle atrophy effects at high substance concentrations has also been observed in previous studies by Chen et al. [35].

In animal experiments, as confirmed in previous studies, weight loss and grip strength were observed in DEX-treated mice [35,36]. Co-treatment with DEX and C-peptide suppressed weight loss and grip strength. Western blotting and RT-qPCR were performed to identify the underlying mechanisms. Analysis of mRNA expression in the Gas muscle showed no significant changes in mRNA expression related to muscle atrophy in the DEX group compared to the CON group. However, the DEX + CP group showed decreased expression of *Klf15* and *Fbxo32* compared to the CON and DEX groups. *Foxo3a* exhibited a decreasing trend in the DEX group compared to the CON group, and significantly decreased in the DEX+CP group compared to both the CON and DEX groups. Additionally, the protein expression level in the Gas muscle exhibited an increase in FoxO3a phosphorylation in the DEX + CP group compared with the DEX group (*p* = 0.0583), and the expression of Atrogin-1 was significantly decreased in the DEX + CP group compared to the other two groups. In the Quad muscle, *Fbxo32* showed a significant decrease in the DEX + CP group compared with the CON group, and exhibited a decreasing trend (*p* = 0.0829) compared with the DEX group. Unlike Gas, *Foxo3a* gene expression did not show differences between groups. The level of protein expression showed a significant increase in the phosphorylation of FoxO3a in the DEX + CP group compared to the DEX group, which had decreased phosphorylation compared to the CON group. Additionally, MuRF1 was significantly reduced in the DEX + CP group compared with the DEX group, Atrogin-1 was significantly reduced in the DEX + CP group compared to that in the other two groups. On the other hand, Gas and Quad muscles did not show significant differences in protein synthesis and autophagy related markers among the groups. These results confirmed that co-treatment with DEX and C-peptide reduced the expression of Atrogin-1 in both muscle tissues and inhibited muscle atrophy.

In conclusion, we observed that C-peptide increased MHC protein expression in dexamethasone-induced atrophy of C2C12 myotubes. Furthermore, in the dexamethasone-induced muscle atrophy mouse model, C-peptide treatment increased muscle mass and strength through reducing the expression of protein degradation markers. Therefore, C-peptide is considered to exert a significant impact on myocyte differentiation and muscle regulation.

A limitation of this study was the inability to maintain C-peptide blood concentrations in vivo owing to its short half-life of 20–30 min, and the use of intraperitoneal injections [20]. Nevertheless, this study clarified the muscle regulatory mechanism through molecular biology analysis in a situation where there is almost no prior research on C-peptide related to muscle regulation, thus confirming the potential of C-peptide as a bioactive substance for controlling muscle atrophy. Furthermore, these findings serve as foundational underpinnings to facilitate the prospective assessment of C-peptide’s efficacy in diabetic muscle atrophy.

## 4. Materials and Methods

### 4.1. Cell Culture and Differentiation

C2C12 myoblast (ATCC, Manassas, VA, USA) were cultured with Dulbecco’s modified Eagle’s medium (DMEM, Welgene, Gyeongsan, Gyeongsangbuk-do, Republic of Korea) supplemented with 10% fetal bovine serum (FBS, Welgene, Republic of Korea), and 1% penicillin–streptomycin (Welgene, Republic of Korea) at 5% CO_2_, 37 °C. For differentiation, confluent cells were exposed to a differentiation medium containing DMEM supplemented with 2% horse serum (HS, Welgene, Republic of Korea). C2C12 myotubes were used in the experiments after 4 days of differentiation. After treatment with 1 μM DEX (D2915, Sigma Aldrich, St. Louis, MO, USA), C-peptide (Human, C-peptide sequence: EAEDLQVGQVELGGGPGAGSLQPLALEGSLQ, molecular formula: C_129_H_211_N_35_O_48_, molecular weight:3020.3, Peptron, Daejeon, Republic of Korea) was added to the differentiated myotubes for 5 h at 0.01, 0.1, 1, 10, 100 nM. C-peptide was diluted in 1 × phosphate-buffered saline (PBS, Bioneer, Daejeon, Republic of Korea).

### 4.2. Immunofluorescence and Determination of the Differentiation and Fusion Indices

C2C12 were treated with or without 1 μM DEX for 5 h, then treated with or without C-peptide (1, 100 nM) for 5 h, washed three times with cold Dulbecco’s phosphate-buffered saline (DPBS, Welgene, Republic of Korea), and incubated with 10% formalin (Sigma Aldrich, USA). After permeabilization by treatment with 0.1% Triton X-100 (Sigma Aldrich, USA) for 10 min, blocking was performed with 3% BSA/PBS at room temperature. C2C12 myotubes were incubated overnight at 4 °C with MHC (MAB4470, R&D Systems, Minneapolis, MN, USA), followed by Alexa 488 (A21202, Thermo Fisher Scientific, Rockford, IL, USA) and 4′,6-diamidino-2-phenylindole (DAPI, D9542, Sigma Aldrich, USA) was incubated at room temperature for 30 min and then washed with DPBS. C2C12 myotube images were obtained using a fluorescence microscope (Motic, San Antonio, TX, USA). The myotube differentiation index was calculated as the ratio of the number of nuclei in MHC-positive myotubes to the total number of nuclei. The ratio of MHC-positive multinucleated tubes of nuclei to the total number of nuclei was expressed as the fusion index.

### 4.3. Animals

Nine-week-old male C57BL/6J mice (Central Lab Animal Inc., Seoul, Republic of Korea) had free access to a water and standard pellet diet and were maintained in a controlled environment (12 h light/dark cycle, 25 ± 2 °C, relative humidity 50–60%). Animal experiments were conducted with the approval of the Ethics Committee of Ewha Womans University (Seoul, Republic of Korea, IACUC 19-046). After one week of adaptation, the mice were randomly divided into three groups, with six mice in each group. Mice in the CON group were intraperitoneally injected with saline for 20 days. In the DEX group, muscle atrophy was induced by injecting 20 mg/kg/day of DEX for 8 days, followed by an intraperitoneal injection of saline for 12 days. The DEX + CP group was injected with 20 mg/kg/day DEX and 50 μg/day C-peptide for 8 days, followed by injection of 50 μg/day C-peptide for 12 days. The body weights of the mice were measured every 4 days from the start of the experiment. To assess skeletal muscle strength, grip strength was measured using a push–pull gauge (FGJN-5; Nidec-Shimpo, Kyoto, Japan). The mouse grabbed the grid with all four limbs, the experimenter held its tail, and pulled it with a constant force until the mouse separated from the grid. When the mouse was separated from the grid, it was measured three times per subject, and the maximum value was used for the analysis. After the experimental period, Gas, Quad, and TA muscles, epididymal fat tissue, and liver were extracted, weighed, and frozen at −70 °C. To measure the concentration of serum C-peptide in the groups, the blood of the experimental animals was collected and processed according to the manufacturer’s protocol using a Mouse C-peptide ELISA kit (Alpco, Salem, NH, USA).

### 4.4. Hematoxylin and Eosin Staining and Determination of the Cross-Sectional Area

To analyze the cross-sectional area of Gas muscle and Quad muscle, muscle tissue fixed with 10% formalin (Sigma Aldrich, USA) was embedded with paraffin, cut into 5 µm-thick slices, and then stained with Hematoxylin and Eosin. The stained muscle sections were magnified by 200× to obtain four representative images per group. CSA was measured using the ImageJ software 1.53t (National Institutes of Health, Bethesda, MD, USA), and more than 700 sets of data were collected per group.

### 4.5. RNA Isolation and Reverse Transcription-Quantitative Polymerase Chain Reaction

RNA was extracted using QIAzol^®^ Lysis Reagent (QIAGEN, Hilden, Germany) to analyze mRNA expression in Gas and Quad muscles. The extracted RNA was quantified with a Nabi-UV/VIS Nano Spectrophotometer (Micro Digital, Seongnam, Gyeonggi-do, Republic of Korea), and complementary DNA was synthesized using TOPscript RT DryMIX kit (Enzynomics, Daejeon, Republic of Korea). Gene expression analysis was performed with the CFX Connect™ Real time system (Bio-rad, Hercules, CA, USA) using TOPreal SYBR Green qPCR PreMIX (Enzynomics, Republic of Korea), according to the manufacturer’s protocol. The PCR cycling was 95 °C for 10 min and 40 cycles of 95 °C for 10 s, 60 °C for 15 s, and 72 °C for 30 s. The results were quantitatively analyzed by the 2^−ΔΔCq^ method, and all genes were standardized using GAPDH. The nucleotide sequences of the primers for the gene were as shown in Table 1.

### 4.6. Protein Isolation and Western Blotting

Proteins from C2C12 cells were extracted using Cell Lysis Buffer (Cell Signaling Technology, Beverly, MA, USA) containing phenylmethylsulfonyl fluoride (PMSF, Thermo Fisher Scientific, USA), a protease inhibitor cocktail (Roche, Mannheim, Germany), and a phosphatase inhibitor cocktail (Roche, Germany). T-PER™ Tissue Protein Extraction Reagent (Thermo Fisher Scientific, USA) was used as a buffer for protein extraction from Gas and Quad muscles. After disruption, the supernatant was collected using a centrifuge (12,000 rpm, 10 min, 4 °C). The concentration of the extracted proteins was quantified using the Quick Start Bradford Protein Assay (Bio-Rad, USA) [37]. Protein samples (10 μg) were separated with SDS-PAGE and transferred to PVDF membranes. The membranes were blocked with 5% skim milk (BD Biosciences, Franklin Lakes, NJ, USA) in Tris-Buffered saline with Tween-20 (TBST, Biosolution, Suwon, Republic of Korea) for 1 h at room temperature, and then incubated with the following primary antibody overnight at 4 °C: GAPDH (Ab8245) and Tubulin (Ab7291), which were obtained from Abcam (Cambridge, UK); p-Akt (9271), Akt (9272), p-FoxO3a (9465), FoxO3a (12829), p-P70 S6K1 (9205), P70 S6K1 (2708), p-4EBP1 (2855), 4EBP1 (9644) and SQSTM1/p62 (39749), purchased from Cell Signaling Technology, and MuRF1(Sc-398608), Atrogin-1(Sc-166806) and KLF15(Sc-271675) sourced from Santa Cruz Biotechnology (Dallas, TX, USA); LC3B(NB100-2220) was procured from Novus Biologicals (Centennial, CO, USA). After washing thrice with TBST, the membranes were incubated with HRP-conjugated anti-mouse IgG or HRP-conjugated anti-rabbit IgG (Cell Signaling Technology). Protein bands were visualized using Westar Sun (Cyanagen, Bologna, Italy) and Azure 300 Imaging Systems (Azure Biosystems, Dublin, CA, USA). Quantitative analysis was performed using the ImageJ software (National Institutes of Health, USA).

### 4.7. Statistical Analyses

GraphPad Prism 9.0.0 software (La Jolla, CA, USA) was used for data visualization and statistical analyses. The results are expressed as the mean ± standard error of the mean. An unpaired Student’s *t*-test was performed used to identify the differences among groups.

## Figures and Tables

**Figure 1 ijms-24-15433-f001:**
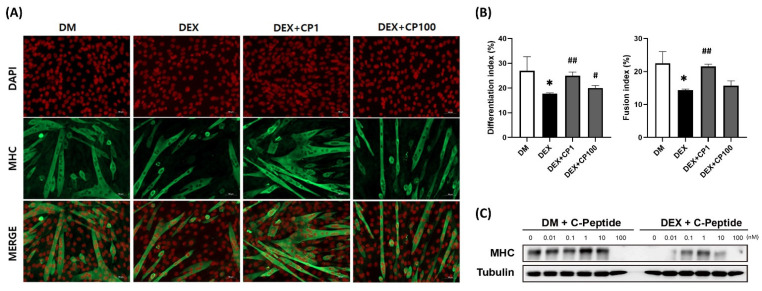
Effect of C-peptide on the differentiation of C2C12 myotubes. (**A**) Immunofluorescence staining of DAPI (red) and MHC (green) in C2C12 myotubes; (**B**) Differentiation and fusion indices of C2C12 myotubes. Differentiation index = number of total nuclei in MHC-positive myotubes/total nuclei. Fusion index = number of multi-nuclei in MHC-positive myotubes/total nuclei. (**C**) Representative image of the MHC Western blot. Data are represented as the mean ± standard error of the mean. Statistical analysis was conducted using an unpaired Student’s *t*-test; DEX vs. DM, * *p* < 0.05; DEX + CP vs. DEX, *^#^ p* < 0.05, ^##^
*p* < 0.01; DM, differentiation media; DEX, dexamethasone; CP, C-peptide; DAPI, 4′,6-diamidino-2-phenylindole; MHC, myosin heavy chain.

**Figure 2 ijms-24-15433-f002:**
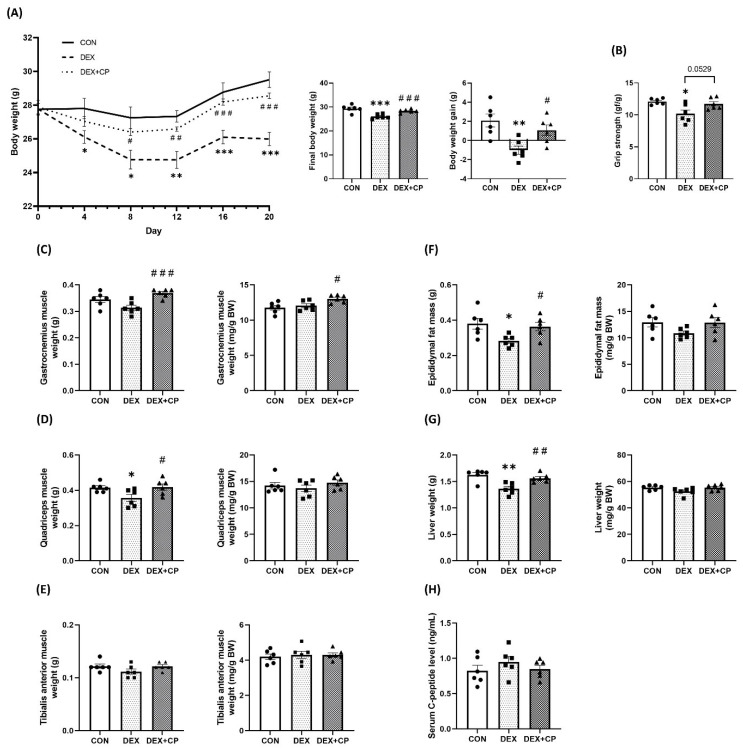
Effects of C-peptide on the body weight, grip strength, skeletal muscle, epididymal fat, and liver weight in C57BL/6J mice, and serum C-peptide levels. (**A**) Body weight. (**B**) Grip strength. (**C**–**E**) Muscle weight. (**F**) Epididymal fat. (**G**) Liver weight. (**H**) C-peptide level in serum. Data are represented as the mean ± standard error of the mean (*n* = 6). Statistical analysis was conducted using an unpaired Student’s *t*-test; DEX vs. CON, * *p* < 0.05, ** *p* < 0.01, *** *p* < 0.001; DEX + CP vs. DEX, *^#^ p* < 0.05, ^##^
*p* < 0.01, ^###^
*p* < 0.001; CON, control group; DEX, dexamethasone group; DEX + CP, dexamethasone + C-peptide group; BW, body weight.

**Figure 3 ijms-24-15433-f003:**
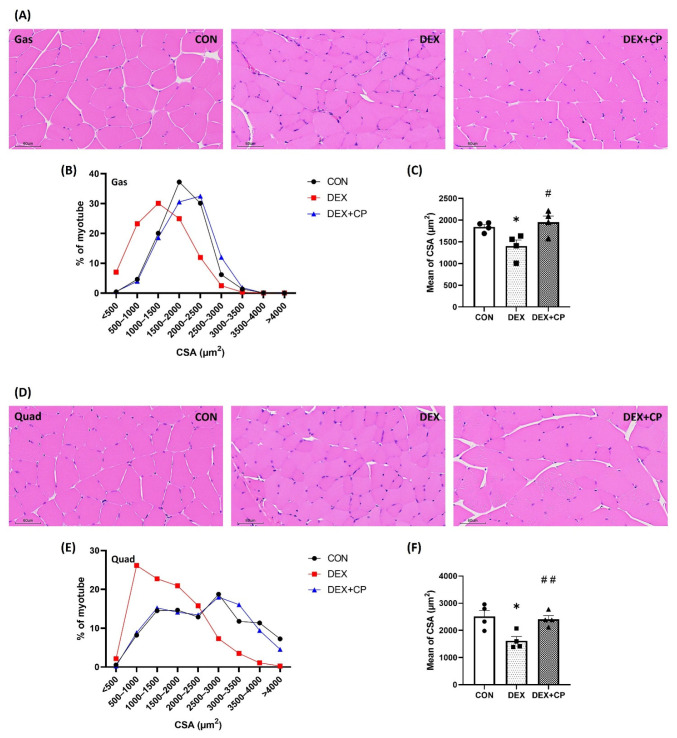
Effect of C-peptide on muscle morphology in C57BL/6J mice. (**A**) Hematoxylin and eosin staining of Gas muscles. (**B**) Distribution of myofiber area of Gas muscles. (**C**) Average of myofiber area of Gas muscles. (**D**) Hematoxylin and eosin staining of Quad muscles. (**E**) Distribution of myofiber area of Quad muscle; (**F**) Average of myofiber area of Quad muscles. Data are represented as the mean ± standard error of the mean (*n* > 700/group). Statistical analysis was conducted using an unpaired Student’s *t*-test; DEX vs. CON, * *p* < 0.05; DEX + CP vs. DEX, *^#^ p* < 0.05, ^##^
*p* < 0.01; Gas, gastrocnemius; Quad, quadriceps; CSA, cross sectional area; CON, control group; DEX, dexamethasone group; DEX + CP, dexamethasone + C-peptide group.

**Figure 4 ijms-24-15433-f004:**
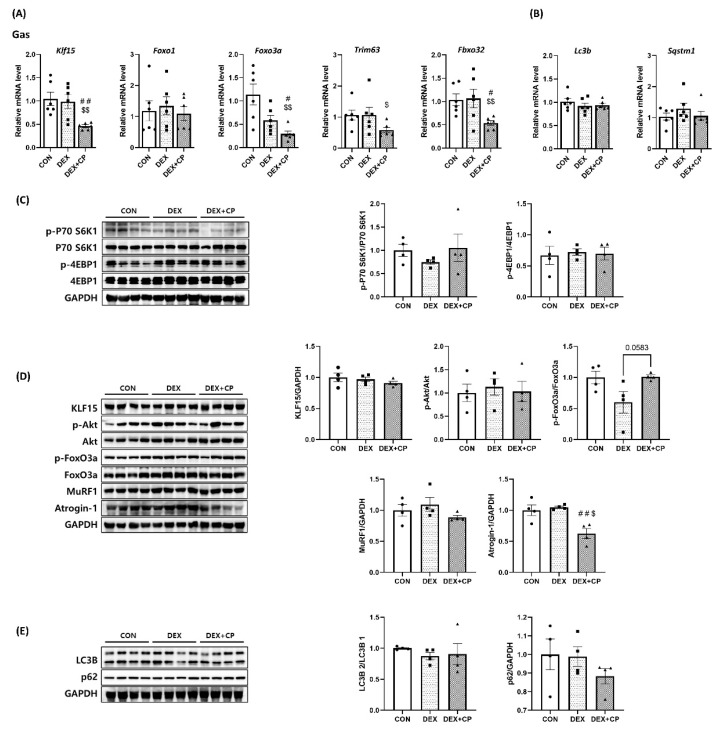
Effects of C-peptide on the mRNA and protein expression levels in gastrocnemius muscles. Related mRNA levels of (**A**) muscle degradation and (**B**) autophagy (*n* = 6). Western blotting images and quantitative analysis of the protein levels related (**C**) muscle synthesis, (**D**) muscle degradation, and (**E**) autophagy (*n* = 4). Data are represented as the mean ± standard error of the mean. Statistical analysis was conducted using an unpaired Student’s *t*-test; DEX + CP vs. DEX, ^#^
*p* < 0.05, ^##^
*p* < 0.01; DEX + CP vs. CON, ^$^
*p* < 0.05, ^$$^
*p* < 0.01; Gas, gastrocnemius; CON, control group; DEX, dexamethasone group; DEX + CP, dexamethasone + C-peptide group.

**Figure 5 ijms-24-15433-f005:**
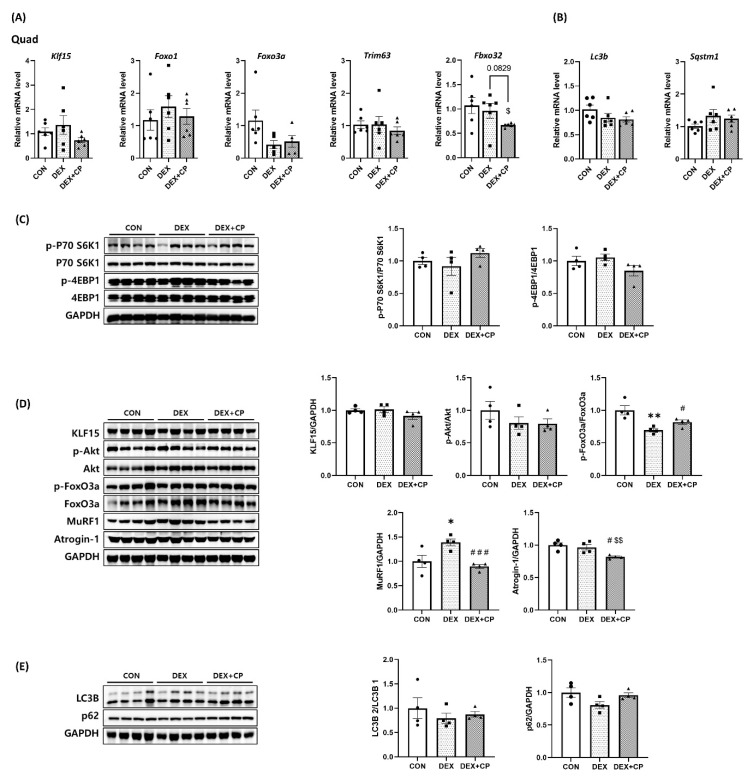
Effects of C-peptide on the mRNA and protein expression levels in quadriceps muscles. Related mRNA levels of (**A**) muscle degradation and (**B**) autophagy (*n* = 6). Western blotting images and quantitative analysis of protein levels related (**C**) muscle synthesis, (**D**) muscle degradation, and (**E**) autophagy (*n* = 4). Data are represented as the mean ± standard error of the mean (*n* = 4). All outliers were identified using the ROUT test. Statistical analysis was conducted using an unpaired Student’s *t*-test; DEX vs. CON, * *p* < 0.05, ** *p* < *0.01*; DEX + CP vs. DEX, *^#^ p* < 0.05, ^###^
*p* < 0.001; DEX + CP vs. CON, ^$^
*p* < 0.05, ^$$^
*p* < 0.01; Quad, quadriceps; CON, control group; DEX, dexamethasone group; DEX + CP, dexamethasone + C-peptide group.

**Table 1 ijms-24-15433-t001:** Primers used for real-time quantitative polymerase chain reaction (RT-qPCR).

Gene	Direction	Sequence (5′–3′)
*Foxo1*	Forward	GTGAACACCAATGCCTCACAC
	Reverse	CACAGTCCAAGCGTCAATA
*Foxo3a*	Forward	AGCCGTGTACTGTGGAGCTT
	Reverse	TCTTGGCGGTATATGGGAAG
*Fbxo32*	Forward	ATGCACACTGGTGCAGAGAG
	Reverse	TGTAAGCACACAGGCAGGTC
*Trim63*	Forward	ACCTGCTGGTGGAAAACATC
	Reverse	CTTCGTGTTCCTTGCACATC
*Klf15*	Forward	GAGACCTTCTCGTCACCGAAA
	Reverse	GCTGGAGACATCGCTGTCAT
*Sqstm1*	Forward	AGGATGGGGACTTGGTTGC
	Reverse	TCACAGATCACATTGGGGTGC
*Lc3b*	Forward	CCCACCAAGATCCCAGTGAT
	Reverse	CCAGGAACTTGGTCTTGTCCA
*Gapdh*	Forward	TCCCACTCTTCCACCTTCGA
	Reverse	CAGGAAATGAGCTTGACAAAGTTG

*Foxo*, forkhead box; *Fbxo32*, F-Box Protein 32; *Trim63*, tripartite motif-containing 63; *Klf15*, Kruppel-like factor 15; *Sqstm1*, sequestosome 1; *Lc3b*, light chain 3 beta; *Gapdh*, glyceraldehyde 3-phosphate dehydrogenase.

## Data Availability

Not applicable.

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
