# Peer review of "Effects of C-Peptide on Dexamethasone-Induced In Vitro and In Vivo Models as a Potential Therapeutic Agent for Muscle Atrophy"

_ijms, 2023, doi:10.3390/ijms242015433_

Round 1

Reviewer 1 Report

Major comment:

Major

In this study, muscle atrophy is artificially induced with Dexamethasone. However, the authors must cite literature to convey that the cell and molecular mechanisms of muscle atrophy with Dexamethasone treatment are similar when caused by "..starvation, aging, immobility, cachexia, acute injuries, and chronic diseases, such as heart failure, chronic kidney disease, and obstructive pulmonary disease".  (or even diabetes)

2.  Authors have not cited other articles that discuss the physiological effects and molecular targets of C-peptide pro insulin on muscle atrophy: example Maurotti et al., 2023; manuscript on C-peptide and diabetic muscle atrophy https://pubmed.ncbi.nlm.nih.gov/36878894/. 

Authors should cite all relevant literature pertaining to C-peptide treatment for muscle atrophy.  

Minor

Figure 1 legend: Can clarify that the DM group is the same as the CON group if that is the case.  

Author Response

Reviewer 1

[COMMENT #1]

In this study, muscle atrophy is artificially induced with Dexamethasone. However, the authors must cite literature to convey that the cell and molecular mechanisms of muscle atrophy with Dexamethasone treatment are similar when caused by "..starvation, aging, immobility, cachexia, acute injuries, and chronic diseases, such as heart failure, chronic kidney disease, and obstructive pulmonary disease".  (or even diabetes)

[RESPONSE #1]

  • Thank you for your kind comments. Following the comment, we inserted sentences and reference No. in manuscript [line 70-71 and 463]

[COMMENT #2]

Authors have not cited other articles that discuss the physiological effects and molecular targets of C-peptide pro insulin on muscle atrophy: example Maurotti et al., 2023; manuscript on C-peptide and diabetic muscle atrophy https://pubmed.ncbi.nlm.nih.gov/36878894/.

Authors should cite all relevant literature pertaining to C-peptide treatment for muscle atrophy.

[RESPONSE #2]

  • Following the comment, we revised sentences and inserted reference No. in manuscript [line 83-85]

[COMMENT #3]

Figure 1 legend: Can clarify that the DM group is the same as the CON group if that is the case.

[RESPONSE #3]

  • Following the comment, we revised “CON → DM” in Figure 1 legend [line 118].

Reviewer 2 Report

The manuscript Effects of C-Peptide on Dexamethasone-Induced In Vitro and In Vivo Models as a Potential Therapeutic Agent for Muscle Atrophyby by Kim et al. describes the combined effect of C-peptide and dexamethasone on muscle atrophy and muscular strength in a mouse model.

The study is well described and includes sound statistics data on….

Abstract:

Lines 16-18, Please, remove words that are used multiple times (treated, treated, treated, treatment).

Line 21, what is CP? It was not abbreviated before.

Line 27, please, rewrite the last sentence to something like ‘In addition, the work presents data on…’.

Introduction:

Line 33, I am not so sure if reference 1 is the best to use for a very general statement on percentages of muscle presence in the human body. Reference 1 is basically focused on muscle importance in healthy and in diseased states.

Line 38, something is missing in this phrase.

Line 48, I would add a reference to Kitajima and colleagues (2020) Journal of physiological sciences.

Line 70, to what proteins or other molecules does C-peptide link proinsulin?

Line 78, maybe explain that C2C12 cells are derived from a myoblast mouse cell line.

Materials and methods:

Line 298, please state which type of Dexamethasone from Sigma you are using. There are several with different catalogue numbers.

Line 334, I have the feeling a few words are missing? The mouse held the grid with four legs, how does it use its tail? Maybe I misunderstand this phrase, so please explain a little more.

Line 360, the authors must be aware of the fact that relative quantitation to only one constitutively expressed gene like GAPDH is not very good practice. Do you have more data so you can derive an average value for the standard expression? So, a set of housekeeping genes?

Line 498, please, as this is the international journal of molecular sciences, do indicate minimally the cycling conditions of your qPCR.

Lines 380 and further: if multiple detection antibodies come from the same supplier, please remove the redundant words for readability and space reasons.

Results:

Line 90, Maybe add the short explanation on fusion index to the text (% MyHC nuclei etc.).

Line 172, Is not it worth mentioning that Fox03A is already significantly reduced in DEX groups compared to the controls? The effects is even increased upon incubation with C peptide. This does not seem to be the case in quad muscles, where combined treatment increases slightly the expression. This point also comes back in the Discussion section, please have a closer look at that (lines 260 and further).

Line 187, word is missing: ‘… MuRF1 levels did not show differences.

Line 188, tended to increase? I would make it stronger: it increased, since it did not tend; you measured it. And it increased compared to the DEX group (compared is missing).

Discussion:

Line 220, Skeletal muscles make up our bodies. This is very generalized and not true, since many tissues ‘make up our bodies’. Please, rewrite the sentence.

Lines 229 and further (whole paragraph), the information on C peptide does belong to the Introduction section, to my opinion. You can refer to that information here.

Line 277, this is a strong conclusion. Maybe put it somewhat more carefully? Augmentation appears to be ascribed to suppression. Yes, there are effects observed in microscopy, western blot and mRNA expression, but especially the latter are significant, though not very strong effects? Can you comment on that?

English language: good. However, some minor issues.

Author Response

Reviewer 2

[COMMENT #1]

Lines 16-18, Please, remove words that are used multiple times (treated, treated, treated, treatment).

[RESPONSE #1]

  • Thank you for your point. Following the comment, we revised abstract [line 15-17].

[COMMENT #2]

Line 21, what is CP? It was not abbreviated before.

[RESPONSE #2]

  • Following the comment, we revised “CP → C-peptide” in abstract [line 21].

[COMMENT #3]

Line 27, please, rewrite the last sentence to something like ‘In addition, the work presents data on…’.

[RESPONSE #3]

  • Following the comment, we revised abstract [line 26-27].

[COMMENT #4]

Line 33, I am not so sure if reference 1 is the best to use for a very general statement on percentages of muscle presence in the human body. Reference 1 is basically focused on muscle importance in healthy and in diseased states.

[RESPONSE #4]

  • Thank you for your point. Following the comment, we revised references [line 432-433].

[COMMENT #5]

Line 38, something is missing in this phrase.

[RESPONSE #5]

  • Following the comment, we revised sentences [line 38-40].

[COMMENT #6]

Line 48, I would add a reference to Kitajima and colleagues (2020) Journal of physiological sciences.

[RESPONSE #6]

  • Following the comment, we inserted reference No. in manuscript [line 447-448].

[COMMENT #7]

Line 70, to what proteins or other molecules does C-peptide link proinsulin?

[RESPONSE #7]

  • Following the comment, we inserted information [line 72-74].

[COMMENT #8]

Line 78, maybe explain that C2C12 cells are derived from a myoblast mouse cell line.

[RESPONSE #8]

  • Following the comment, we inserted information [line 88-89].

[COMMENT #9]

Line 298, please state which type of Dexamethasone from Sigma you are using. There are several with different catalogue numbers.

[RESPONSE #9]

  • Following the comment, we added information about dexamethasone [line 308-309].

[COMMENT #10]

Line 334, I have the feeling a few words are missing? The mouse held the grid with four legs, how does it use its tail? Maybe I misunderstand this phrase, so please explain a little more.

[RESPONSE #10]

  • Thank you for your kind comments. Following the comment, we inserted information about the grip strength method [line 344-346].

[COMMENT #11]

Line 360, the authors must be aware of the fact that relative quantitation to only one constitutively expressed gene like GAPDH is not very good practice. Do you have more data so you can derive an average value for the standard expression? So, a set of housekeeping genes?

[RESPONSE #11]

  • Thank you for your point. In this study, we performed RT-qPCR using GAPDH and β-actin as a housekeeping gene. However, in our experiment, more stable results were obtained when GAPDH was used.
  • The following references, published in International Journal of Molecular Sciences, used GAPDH as a housekeeping gene.
  1. Webster, J. M.; Sagmeister, M. S.; Fenton, C. G.; Seabright, A. P.; Lai, Y.-C.; Jones, S. W.; Filer, A.; Cooper, M. S.; Lavery, G. G.; Raza, K., Global deletion of 11β-HSD1 prevents muscle wasting associated with glucocorticoid therapy in polyarthritis. J. Mol. Sci. 2021, 22, (15), 7828.
  2. Langendorf, E. K.; Rommens, P. M.; Drees, P.; Mattyasovszky, S. G.; Ritz, U., Detecting the effects of the glucocorticoid dexamethasone on primary human skeletal muscle cells—differences to the murine cell line J. Mol. Sci. 2020, 21, (7), 2497.
  3. Kim, K. W.; Baek, M.-O.; Choi, J.-Y.; Son, K. H.; Yoon, M.-S., Analysis of the molecular signaling signatures of muscle protein wasting between the intercostal muscles and the gastrocnemius muscles in db/db mice. J. Mol. Sci. 2019, 20, (23), 6062.
  4. Liu, Y.; Yang, X.; Jing, X.; He, X.; Wang, L.; Liu, Y.; Liu, D., Transcriptomics analysis on excellent meat quality traits of skeletal muscles of the Chinese indigenous min pig compared with the large white breed. J. Mol. Sci. 2017, 19, (1), 21.

[COMMENT #12]

Line 498, please, as this is the international journal of molecular sciences, do indicate minimally the cycling conditions of your qPCR.

[RESPONSE #12]

  • Following the comment, we inserted additional information in material and methods part [line 369-370].

[COMMENT #13]

Lines 380 and further: if multiple detection antibodies come from the same supplier, please remove the redundant words for readability and space reasons.

[RESPONSE #13]

  • Following the comment, we revised sentence [line 392-397].

[COMMENT #14]

Line 90, Maybe add the short explanation on fusion index to the text (% MyHC nuclei etc.).

[RESPONSE #14]

  • Following the comment, we inserted information about the fusion index [line 100-102].

[COMMENT #15]

Line 172, Is not it worth mentioning that Fox03A is already significantly reduced in DEX groups compared to the controls? The effects is even increased upon incubation with C peptide. This does not seem to be the case in quad muscles, where combined treatment increases slightly the expression. This point also comes back in the Discussion section, please have a closer look at that (lines 260 and further).

[RESPONSE #15]

  • According to reviewer’s comments, decrease of Foxo3a in DEX group compared to CON group was not significant, but we revised the results and discussion were. [line 186-187, 268-270 and 276-277].

[COMMENT #16]

Line 187, word is missing: ‘… MuRF1 levels did not show differences.

[RESPONSE #16]

  • Following the comment, we revised sentence [line 200].

[COMMENT #17]

Line 188, tended to increase? I would make it stronger: it increased, since it did not tend; you measured it. And it increased compared to the DEX group (compared is missing).

[RESPONSE #17]

  • Following the comment, we revised sentence [line 201 and 271-272].

[COMMENT #18]

Line 220, Skeletal muscles make up our bodies. This is very generalized and not true, since many tissues ‘make up our bodies’. Please, rewrite the sentence.

[RESPONSE #18]

  • Thank you for your point. Following the comment, we revised Discussion section [line 233-234].

[COMMENT #19]

Lines 229 and further (whole paragraph), the information on C peptide does belong to the Introduction section, to my opinion. You can refer to that information here.

[RESPONSE #19]

  • Thank you for your kind comments. Following the comment, we revised paragraph [line 77-83].

[COMMENT #20]

Line 277, this is a strong conclusion. Maybe put it somewhat more carefully? Augmentation appears to be ascribed to suppression. Yes, there are effects observed in microscopy, western blot and mRNA expression, but especially the latter are significant, though not very strong effects? Can you comment on that?

[RESPONSE #20]

  • Thank you for your point. According to reviewer’s comments, we revised Discussion [line 286-291].
